# Patch-level Neighborhood Interpolation: A General and Effective Graph-based Regularization Strategy

## Abstract

Regularization plays a crucial role in machine learning models, especially for deep neural networks. The existing regularization techniques mainly rely on the i.i.d. assumption and only consider the knowledge from the current sample, without the leverage of neighboring relationship between samples. In this work, we propose a general regularizer called Patch-level Neighborhood Interpolation (**Pani**) that conducts a non-local representation in the computation of network. Our proposal explicitly constructs patch-level graphs in different network layers and then linearly interpolates neighborhood patch features, serving as a general and effective regularization strategy. Further, we customize our approach into two kinds of popular regularization methods, namely Virtual Adversarial Training (VAT) and MixUp as well as its variants. The first derived **Pani VAT** presents a novel way to construct non-local adversarial smoothness by employing patch-level interpolated perturbations. In addition, the second derived **Pani MixUp** method extends the original MixUp regularization and its variant to the Pani version, achieving a significant improvement in the performance. Finally, extensive experiments are conducted to verify the effectiveness of our Patch-level Neighborhood Interpolation approach in both supervised and semi-supervised settings.

## 1 Introduction

In the statistical learning theory, regularization techniques are typically leveraged to achieve the trade-off between empirical error minimization and the control of model complexity (Vapnik & Chervonenkis, 2015). In contrast to the classical convex empirical risk minimization where regularization can rule out trivial solutions, regularization plays a rather different role in deep learning due to its highly non-convex optimization nature (Zhang et al., 2016). Among all the explicit and implicit regularization, regularization with stochastic transformation, perturbations and randomness, such as adversarial training (Goodfellow et al., 2014), dropout and MixUp (Zhang et al., 2017), play a key role in the deep learning models due to their superiority in the performance (Berthelot et al., 2019b; Zhang et al., 2017; Miyato et al., 2018; Berthelot et al., 2019a). In this section, we firstly review two kinds of effective and prestigious regularization branches for deep neural networks, which can elegantly generalize from supervised learning to semi-supervised setting.

Adversarial Training (Goodfellow et al., 2014; Madry et al., 2017) can provide an additional regularization beyond that provided by other generic regularization strategies, such as dropout, pretraining and model averaging. However, recent works (Zhang et al., 2019; Tsipras et al., 2018) demonstrated that this kind of training method holds a trade-off between the robustness and accuracy, limiting the efficacy of the adversarial regularization. Besides, Virtual Adversarial Training (VAT) (Miyato et al., 2018) can be regarded as a natural extension of adversarial training to semi-supervised setting through adversarially smoothing the posterior output distribution with the leverage of unlabeled data. This strategy has achieved great success in image classification (Miyato et al., 2018), text classification (Miyato et al., 2016) and node classification (Sun et al., 2019). Tangent-Normal Adversarial Regularization (TNAR) (Yu et al., 2019) extended VAT by taking the data manifold into consideration and applied VAT along the tangent space and the orthogonal normal space of the data manifold, outperforming previous semi-supervised approaches.

MixUp (Zhang et al., 2017) augmented the training data by incorporating the prior knowledge that linear interpolation of input vectors should lead to linear interpolation of the associated targets, accomplishing consistent improvement of generalization on image, speech and tabular data. Mix-Match (Berthelot et al., 2019b) extended MixUp to semi-supervised tasks by guessing low-entropy labels for data-augmented unlabeled examples and mixing labeled and unlabeled data using MixUp. In contrast with VAT, MixMatch (Berthelot et al., 2019b) utilizes one specific form of consistency regularization, i.e., using the standard data augmentation for images, such as random horizontal flips, rather than computing adversarial perturbations to smooth the posterior distribution of the classifier.

Nevertheless, the vast majority of regularization methods, including the aforementioned approaches, assume that the training samples are drawn independently and identically from an unknown data generating distribution. For instance, Support Vector Machine (SVM), Back-Propagation (BP) for Neural Networks, and many other common algorithms implicitly make this assumption as part of their derivation. However, this i.i.d. assumption is commonly violated in realistic scenarios where batches or sub-groups of training samples are likely to have internal correlations. In particular, Dundar et al. (2007) demonstrated that accounting for the correlations in real-world training data leads to statistically significant improvements in accuracy. Similarly, Peer-Regularized Networks (Peer-Net) (Svoboda et al., 2018) applied graph convolutions (Velickovic et al., 2017; Kipf & Welling, 2016) to harness information of peer samples, and verified its effectiveness on defending adversarial attacks. Motivated by these facts, we aim to design a general regularization strategy that can fully utilize the internal relationship between samples by explicitly constructing a graph within a mini-batch in order to consistently improve the generalization of deep neural networks in both semi- and supervised settings.

In this paper, we propose the Patch-level Neighborhood Interpolation (**Pani**) for deep neural networks, serving as a simple yet effective non-local regularization. We firstly construct a patch-level graph in each mini-batch during the *stochastic gradient decent* training process. Then we apply linear interpolation on the neighboring patch features and the resulting non-local representation additionally captures the relationship of neighboring patch features in different layers, serving as a general and effective regularization. Furthermore, to prove the generality and superiority of our Pani method, we explicitly customize our approach into two kinds of popular and general regularization strategies, i.e., Virtual Adversarial Regularization and MixUp, resulting in **Pani VAT** and **Pani MixUp**. For the Pani VAT, we reformulate the construction of adversarial perturbations, transforming from solely depending on the current sample to the linear interpolation of neighboring patch features. This non-local adversarial perturbations can leverage the information of neighboring correlation from all samples within a batch, providing more informative adversarial smoothness in semi-supervised setting. Besides, in the Pani MixUp, we extend MixUp and its variant MixMatch from image to patch level by mixing fine-grained patch features and corresponding supervised signals. Finally, we conduct extensive experiments to demonstrate that both of the two derived regularization strategies can outperform other state-of-the-art approaches in both supervised and semi-supervised tasks. More importantly, these successful cases verify the generality and superiority of our Patch-level Neighborhood Interpolation method. Our contributions can be summarized as follow:

- We propose a general interpolation strategy either in input or feature space, i.e., Patch-level Neighborhood Interpolation, helping to improve the generalization of deep neural networks on both semi- and supervised scenarios. This strategy can serve as an effective graph-based representation method and has much potential to be leveraged in a wider range of tasks.

- Based on our method, the customized approaches Pani VAT and Pani MixUP as well as Pani MixMatch can boost the generalization performance significantly, and thus provide a guidance to the deployment of our Pani strategy into more regularization methods.

## 2  OUR METHOD: PATCH-LEVEL NEIGHBORHOOD INTERPOLATION

Before introducing our approach, we highly recommend readers to go through some preliminary knowledge about VAT (Miyato et al., 2017), MixUP (Zhang et al., 2017) and PeerNet (Svoboda et al., 2018) in Appendix A. For our method, one related work is PeerNet (Svoboda et al., 2018) that designed graph-based layers to defend against adversarial attacks, but unfortunately the construction of pixel-level $K$-NN graphs in PeerNet (Svoboda et al., 2018) is costly in computation. By contrast, our motivation is to develop a general regularization that can consistently boost the performance of

deep neural networks in both semi- and supervised settings rather than the adversarial scenario. Besides, the construction way of a non-local layer in our method is more flexible and can be determined by the specific objective function, as elaborated in Section 2.1 and 2.2. Moreover, our patch-level method can achieve computational advantage over pixel-level regularization, and incorporates more meaningful semantic correlations in different layers. Particularly, a flexible patch size can be chosen according to the size of receptive field in different layers, yielding more informative graph-based representation and better regularization performance.

Concretely, as our Patch-level Neighborhood Interpolation (**Pani**) shown in Figure 1, in the first step we determine the candidate peer images set $\mathcal{S}_i$ for each image $i$. This can be achieved by random matching or computing the semantically nearest image neighbors using e.g. the cosine distance. Next, we construct the whole patches set $\mathcal{P}_i$ on the candidate peer images set $\mathcal{S}_i$ for each image $i$ by clipping the corresponding patches in the different locations on an input or a feature map. Following the establishment of patch set $\mathcal{P}_i$, we construct $K$ nearest neighbor patch graphs based on the distance of patch features in order to find the neighbors of each patch in patch set $\mathcal{P}_i$ for $\forall i = 1, .., N$. Mathematically, following the definition in the PeerNet, let $\mathbf{z}_p^i$ be the $p$-th patch on the input or feature map $\mathbf{Z}^i$ for the $i$-th image within one batch. Then denote the $k$-th nearest patch neighbor for $\mathbf{z}_p^i$ as $\mathbf{z}_{q_k}^{j_k}$ taken from the patch $q_k$ of the peer image $j_k$ in the candidate set $\mathcal{S}_i$.

Next, in order to leverage the knowledge from neighbors, different from graph attention mechanism in PeerNet, we apply a more straightforward linear interpolation on the neighboring patches for the current patch $\mathbf{z}_p^i$. Then, the general formulation of our Patch-level Neighborhood Interpolation can be presented as follows:

$$\tilde{\mathbf{z}}_p^i = \mathbf{z}_p^i + \sum_{k=1}^{K} \eta_{ipk}(\mathbf{z}_{q_k}^{j_k} - \mathbf{z}_p^i), \tag{1}$$

where $\eta_{ipk}$ is the combination coefficient for the $p$-th patch of $i$-th image w.r.t its $k$-th patch neighbor, which can be computed through the power iteration similar to the manner of VAT, or through random sampling from a specific distribution in randomness-based regularization, e.g., Mixup and its variants. Moreover, the choice of linear interpolation in Eq. 1 enjoys great computational advantage over the nonlinear GAT form in PeerNet in the computation of networks. Finally, after the patch-level linear interpolation on patch features, we can obtain the refined graph-based representation $\tilde{\mathbf{Z}}^i$ for $i$-th image, $\forall i = 1, ..., N$.

Note that our proposed method can explicitly combine the advantages of manifold regularization and non-local filtering in a flexible way, which we have a more detailed discussion about in Appendix B. Besides, to further demonstrate the generality and effectiveness of our Pani method, we provide Pani-version of two typical regularization strategies, i.e., Virtual adversarial Training and Mixup as

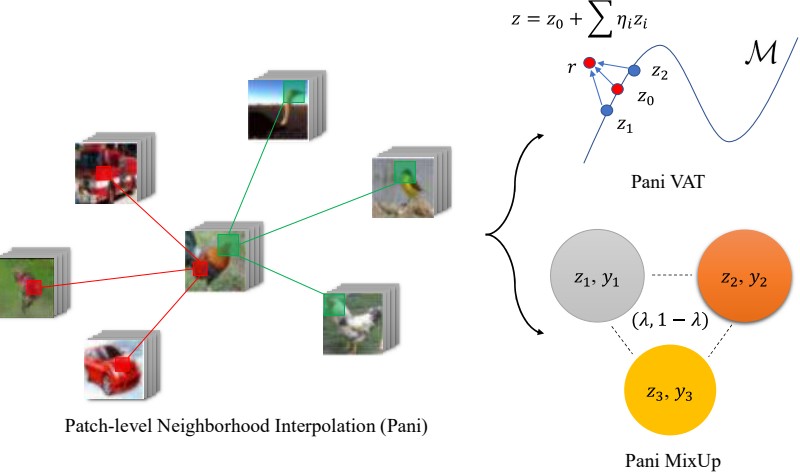

Figure 1: Pipeline of our Patch-level Neighborhood Interpolation followed by two derived regularization, i.e., Pani VAT and Pani MixUp. $r$ represents the perturbation constructed by our method and $(\lambda, 1 - \lambda)$ is the mixing coefficient pair.

well its variant MixMatch, and verify the superiority of our Pani strategy on the significant boosting of accuracy.

## 2.1 PANI VAT

Based on our Patch-level Neighborhood Interpolation framework, we can construct a novel Pani VAT that utilizes the combination or interpolation of patch neighbors for each sample to manipulate the non-local perturbations, thus providing more informative adversarial smoothness in semi-supervised setting. Consider a more general composite function form of the classifier $f$, i.e., $f(x) = g(z)$ and $z = h(x)$ where $z$ denotes the hidden feature of input $x$ or the input itself when the reduced form happens. Combining VAT formulation, i.e., Eq. 7 in Appendix A, and Pani formulation, i.e., Eq. 1, we reformulate our Pani VAT with perturbations on $L$ layers in a deep neural network as follows:

$$\max_{\eta} D[g(z), g(\tilde{z}(\eta))]$$
$$s.t. \ \sum_{l=1}^{L} w_l^2 \|\eta^{(l)}\|^2 \le \epsilon^2, \tag{2}$$

where $D$ measures the divergence between two distributions. $\eta = \{\eta_{ipk}\}$ denotes the generic perturbations from our Pani method and $\eta^{(l)}$ indicates the perturbations in $l$-th layer of network. $\tilde{z}(\eta) = \{\tilde{\mathbf{z}}_p^i\}$ represents the smoothed feature map imposed by perturbation $\eta$ considering all patches in the way shown in Eq. 1. In particular, when $L = 1$, adversarial perturbations are only imposed on the input feature, which is similar to the traditional virtual adversarial perturbations. Additionally, $w_l$ is the hyper-parameter, adjusting the weight of perturbation $\eta^{(l)}$ in different layers with the overall perturbations restrained in an $\epsilon$-ball.

Next, we still utilize the similar power iteration and finite difference proposed in VAT (Miyato et al., 2017) to compute the desired perturbation $\eta^*$. Then the resulting full loss function is defined as:

$$\min_{\theta} \mathcal{L}_0 + \beta \mathbb{E}_{x \sim \mathcal{D}} \mathcal{R}_{\text{vadv}}(x, \eta^*; \theta), \tag{3}$$

where $\mathcal{L}_0$ is the original supervised loss and $\beta$ controls the degree of adversarial smoothness. $\mathcal{R}_{\text{vadv}}(x, y, \eta^*) = D[g(z), g(\tilde{z}(\eta^*))]$ can be attained after solving the optimization problem in Eq. 2. For the implementation details, we describe them in Algorithm 1.

---

**Algorithm 1** : Pani VAT within a Batch
1: **Input:** neighbors $K_1$ and $K_2$, classifier $f$, batch size $B$, perturbed layers $L$
2: **Initialization:** combination coefficient $\eta$
3: Compute $K_1$ nearest image neighbors based on the distance of the second last layer output from $f$ and obtain $K_1$ ($K_1 \le B$) peer images set $\mathcal{S}_i$ for each image $i$.
4: **for** $l = 1$ **to** $L$ **do**:
5:      Compute the patch set $\mathcal{P}_i$ for all $K_1$ peer images on layer $l$ for each image $i$ .
6:      Construct a $K_2$ nearest patch neighbors graph for each patch in each image $i$.
7:      Conduct Patch-level Neighborhood Interpolation via Eq. 1 for each patch.
8: **end for**
9: Conduct power iteration and finite difference in VAT to compute $\eta^*$ constrained by Eq. 2.
10: **Return** $\mathcal{R}_{\text{vadv}}(x, \eta^*; \theta)$

---

**Remark.** As shown in the adversarial part of Figure 1, the rationality of our Pani VAT method lies in the fact that the constructed perturbations can entail more non-local information coming from the neighbors of current sample. Through the delicate patch-level interpolation among neighbors of each patch, the resulting non-local virtual adversarial perturbations are expected to provide more informative smoothness, thus enhancing the performance of classifier in the semi-supervised setting.

## 2.2 PANI MIXUP

Next, we leverage Patch-level Neighborhood Interpolation to derive Pani MixUp. The core formulation of Pani MixUp can be written as:

$$\tilde{\mathbf{z}}_p^i = (1 - \sum_{k=1}^K \eta_{ipk})\mathbf{z}_p^i + \sum_{k=1}^K \eta_{ipk}\mathbf{z}_{q_k}^{j_k}$$

$$\tilde{y}_i = (1 - \sum_{k=1}^K \sum_{p=1}^P \frac{\eta_{ipk}}{P})y_i + \sum_{k=1}^K \sum_{p=1}^P \frac{\eta_{ipk}}{P} y_{j_k}, \tag{4}$$

$$s.t.\ \lambda = 1 - \sum_{k=1}^K \sum_{p=1}^P \frac{\eta_{ipk}}{P},$$

where $(\mathbf{z}^i, y^i)$ are the feature-target pairs randomly drawn from the training data. $P$ is the number of patches in each image and $\lambda \sim \text{Beta}(a, b)$ represents the importance of the current input or target while conducting MixUp. To compute $\eta_{ipk}$, we firstly sample $\lambda$ from $\text{Beta}(a, b)$ and $\eta_{ipk}^0$ from a uniform distribution respectively, then we normalize $\eta_{ipk}^0$ according to the ratio of $\lambda$ to satisfy the constraint in Eq. 4 and thus obtain $\eta_{ipk}$. It should be noted that due to the unsymmetric property of $\lambda$ in our framework, we should tune both $a$ and $b$ in our experiments. For simplicity, we fix $b = 1$ and only consider the $a$ as the hyper-parameter to pay more attention to the importance of current patch, which is inspired by the similar approach in MixMatch (Berthelot et al., 2019b). Here we reformulate Eq. 4 to illustrate that Pani MixUp is naturally derived from our Pani framework through additionally considering several constraints:

$$\tilde{\mathbf{z}}_p^i = \mathbf{z}_p^i + \sum_{k=1}^K \eta_{ipk}(\mathbf{z}_{q_k}^{j_k} - \mathbf{z}_p^i)$$

$$s.t.\ \lambda = 1 - \sum_{k=1}^K \sum_{p=1}^P \frac{\eta_{ipk}}{P}, \forall i = 1, ..., N \tag{5}$$

$$\lambda \sim \text{Beta}(a, b), \quad \eta_{ipk} \in [0, 1], \forall i, p, k$$

where the first constraint in Eq. 5 can be achieved through normalization via $\lambda$. Meanwhile, we impose $\eta_{ipk} \in [0, 1]$ as $\eta_{ipk}$ represents the interpolation coefficient. Further, we elaborate the procedure of Pani MixUp in Algorithm 2.

---

**Algorithm 2** : Pani MixUp within a Batch

---

1: **Input:** neighbors $K$, classifier $f$, batch size $B$, perturbed layers $L$, parameter $a$
2: Compute peer images by random matching and obtain peer images set $\mathcal{S}_i$ for each image $i$.
3: **for** $l = 1$ **to** $L$ **do**:
4:     Compute the patch set $\mathcal{P}_i$ on layer $l$ for each image $i$ .
5:     Construct a $K$ nearest patch neighbors graph for each patch in each image $i$.
6:     Sample initial coefficient $\eta_0^{(l)} = \{\eta_{ipk}^0\}$ from $U(0, 1)$ and $\lambda$ from $\text{Beta}(a, 1)$.
7:     Normalize $\eta_0^{(l)}$ according to the ratio $\lambda$ via Eq. 5 to compute $\eta^{(l)}$.
8:     Conduct Pani MixUp over patch features and labels via Eq. 5 for each patch.
9: **end for**
10: **Return** supervised loss based on mixed features and labels.

---

**Remark.** Different from the role of $\eta$ in the aforementioned Pani VAT where $\eta$ serves as the interpolated perturbations, the physical meaning of $\eta$ in our Pani MixUp approach is the linear interpolation coefficient to conduct MixUp. Despite this distinction, both of the two extended regularization methods are naturally derived from our Patch-level Neighborhood Interpolation framework, further demonstrating the generality and superiority of our Pani strategy.

## 3 EXPERIMENTS

In this section, we conduct extensive experiments for Pani VAT and Pani MixUp and its variant Pani MixMatch on both semi- and supervised settings.

### 3.1 PANI VAT

**Implementation Details.** For fair comparison with VAT and its variants, e.g., VAT + SNTG (Luo et al., 2017) and TNAR (Yu et al., 2019), we choose the standard large convolutional network as the classifier as in (Miyato et al., 2018). For the option of dataset, we focus on the standard semi-supervised setting on CIFAR-10 with 4,000 labeled data. Unless otherwise noted, all the experimental settings in our method are the identical with those in the Vanilla VAT (Miyato et al., 2018). In particular, we conduct our Pani VAT on input layer and one additional hidden layer, yielding two variants Pani VAT (input) and Pani VAT (+hidden). More details can refer to Appendix C.

| Method | CIFAR-10(4,000 labels) |
|---|---|
| VAT (Miyato et al., 2017) | $13.15 \pm 0.2$ |
| VAT + SNTG (Luo et al., 2017) | $12.49 \pm 0.36$ |
| Π model (Laine & Aila, 2016) | $16.55 \pm 0.29$ |
| Mean Teacher (Tarvainen & Valpola, 2017) | $17.74 \pm 0.30$ |
| CCLP (Kamnitsas et al., 2018) | $18.57 \pm 0.41$ |
| ALI (Dumoulin et al., 2016) | $17.99 \pm 1.62$ |
| Improved GAN (Salimans et al., 2016) | $18.63 \pm 2.32$ |
| Tripple GAN (Li et al., 2017) | $16.99 \pm 0.36$ |
| Bad GAN (Dai et al., 2017) | $14.41 \pm 0.30$ |
| LGAN (Qi et al., 2018) | $14.23 \pm 0.27$ |
| Improved GAN + JacobRegu + tangent (Kumar et al., 2017) | $16.20 \pm 1.60$ |
| Improved GAN + ManiReg (Lecouat et al., 2018) | $14.45 \pm 0.21$ |
| TNAR (with generative models) (Yu et al., 2019) | $12.06 \pm 0.35$ |
| Pani VAT (input) | $12.33 \pm 0.091$ |
| Pani VAT (+hidden) | $\mathbf{11.98 \pm 0.106}$ |

Table 1: Classification errors (%) of compared methods on CIFAR-10 dataset without data augmentation. The results of our Pani methods are the average ones under 4 runs.

**Our Results.** Table 1 presents the state-of-the-art performance achieved by Pani VAT (+hidden) compared with other baselines on CIFAR-10. We focus on the baseline methods especially along the direction of variants of VAT and refer to the results from TNAR (with generative models) method (Yu et al., 2019), the previous state-of-the-art variant of VAT that additionally leverages the data manifold by generative models to decompose the directions of virtual adversarial smoothness. It is worthy of remarking that the performance of relevant GAN-based approaches, such as Localized GAN (LGAN) (Qi et al., 2018) as well as TNAR (with generative models) in Table 1, *heavily rely on the established data manifold by the generative models*. It is well-known that one might come across practical difficulties while implementing and deploying these generative models. By contrast, *without the requirement of generative models*, our approach can eliminate this disturbance and can still outperform these baselines. In addition, our Pani VAT (+hidden) achieves slight improvement compared with Pani VAT (input), which serves as an ablation study, and thus verifies the superiority of manifold regularization mentioned in our Pani framework part. Overall, the desirable flexibility along with desirable stability (lower standard deviation shown in Table 1) of our Pani VAT further demonstrates the effectiveness of our Pani strategy.

**Analysis of Computational Cost.** Another noticeable advantage of our approach is the negligible increase of computation cost compared with Vanilla VAT. In particular, one crucial operation in our approach is the construction of patch set $\mathcal{P}$ and it can be accomplished efficiently by $as.strided$ function in Python or through the specific convolution operation in Pytorch or TensorFlow. Additionally, the index of $K$ nearest neighbor graph can be efficiently attained through $topk$ operation. We conduct further

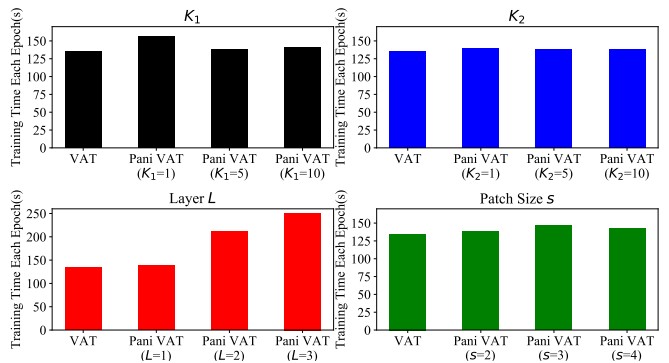

Figure 2: Average training time each epoch with respect to parameters $K_1$, $K_2$, number of layers $L$ and patch size.

sensitivity analysis on the computational cost of our method with respect to other parameters, i.e., $K_1$ (number of peer images), $K_2$ (number of patch neighbors), $L$ (number of perturbed layers) and patch size $s$.

As shown in Figure 2, the variation of all parameters has negligible impact on the training time each epoch compared with Vanilla VAT except the number of perturbed layers. The increasing of computational cost presents an almost linear tendency with the increasing of the number of perturbed layer as the amount of floating-point calculation is proportional to the number of perturbation elements, i.e., $\eta$, if we temporarily neglect the difference of time in the back propagation process for different layers. Combining results from Table 1 and Figure 2, we argue that the better performance can be expected if we construct perturbations on more hidden layers despite the increase of computation.

## 3.2 PANI MIXUP

**Implementation Details** The experimental settings in this section are strictly followed by those in Vanilla MixUp (Zhang et al., 2017) and Vanilla MixMatch (Berthelot et al., 2019b) to pursue fair comparison on CIFAR-10, CIFAR-100 and TinyImageNet datasets. In particular, we compare ERM (Empirical Risk Minimization), MixUp training and our approach for different neural architectures. For fair comparison with input MixUp, we conduct our approach only on input layer and the better performance can be expected naturally if we consider more layers. Besides, we introduce mask mechanism on $\eta$ to avoid overfitting. More details can refer to Appendix C.

| Dataset | Model | Aug | ERM | Mixup($a=1$) | Ours(input) |
|---|---|---|---|---|---|
| CIFAR-10 | PreAct ResNet-18 | ✓ | $5.43 \pm 0.16$ | $4.24 \pm 0.16$ | $\mathbf{3.93 \pm 0.12}$ |
| | | × | $12.81 \pm 0.46$ | $9.88 \pm 0.25$ | $\mathbf{8.12 \pm 0.09}$ |
| | PreActResNet-34 | ✓ | $5.15 \pm 0.12$ | $3.72 \pm 0.20$ | $\mathbf{3.36 \pm 0.15}$ |
| | | × | $12.67 \pm 0.26$ | $10.60 \pm 0.57$ | $\mathbf{8.13 \pm 0.32}$ |
| | WideResNet-28-10 | ✓ | $4.59 \pm 0.06$ | $3.21 \pm 0.13$ | $\mathbf{3.02 \pm 0.11}$ |
| | | × | $8.78 \pm 0.20$ | $8.08 \pm 0.39$ | $\mathbf{5.79 \pm 0.03}$ |
| CIFAR-100 | PreAct ResNet-18 | ✓ | $24.96 \pm 0.51$ | $22.15 \pm 0.72$ | $\mathbf{20.90 \pm 0.21}$ |
| | | × | $39.64 \pm 0.65$ | $41.96 \pm 0.27$ | $\mathbf{32.03 \pm 0.34}$ |
| | PreActResNet-34 | ✓ | $24.85 \pm 0.14$ | $21.49 \pm 0.68$ | $\mathbf{19.46 \pm 0.29}$ |
| | | × | $39.41 \pm 0.80$ | $41.96 \pm 0.24$ | $\mathbf{34.48 \pm 0.86}$ |
| | WideResNet-28-10 | ✓ | $21.00 \pm 0.09$ | $18.58 \pm 0.16$ | $\mathbf{17.39 \pm 0.16}$ |
| | | × | $31.91 \pm 0.77$ | $35.16 \pm 0.33$ | $\mathbf{27.71 \pm 0.63}$ |
| TinyImageNet | PreAct ResNet-18 | ✓ | $44.90 \pm 0.28$ | $42.84 \pm 0.35$ | $\mathbf{42.20 \pm 0.39}$ |
| | | × | $54.95 \pm 0.63$ | $60.58 \pm 0.83$ | $\mathbf{52.25 \pm 0.75}$ |
| | PreActResNet-34 | ✓ | $40.66 \pm 1.64$ | $43.18 \pm 0.84$ | $\mathbf{40.03 \pm 0.61}$ |
| | | × | $51.03 \pm 0.57$ | $55.91 \pm 1.09$ | $\mathbf{49.56 \pm 0.94}$ |
| | WideResNet-28-10 | ✓ | $42.30 \pm 0.51$ | $40.64 \pm 0.77$ | $\mathbf{38.97 \pm 0.81}$ |
| | | × | $48.47 \pm 0.24$ | $51.19 \pm 1.19$ | $\mathbf{46.26 \pm 0.70}$ |

Table 2: Test error in comparison with ERM, MixUp and Pani MixUp (input) across three deep neural network architectures with and without data augmentation. All results are the average ones under 5 runs. Results of MixUp on the settings without data augmentation are based on our implementation on the original code from MixUp.

**Our Results.** Table 2 presents the consistent superiority of Pani MixUp over ERM (normal training) as well as Vanilla MixUp over different deep neural network architectures. It is worthy of noting that the superiority of our approach in the setting without data augmentation can be more easily observed than that with data augmentation. Another interesting phenomenon is that MixUp suffers from one kind of collapse on the performance as the accuracy of MixUp is even inferior to the ERM on CIFAR-100 and TinyImageNet on the setting without data augmentation. By contrast, our approach exhibits consistent advantage ERM and MixUp across various settings and network architectures.

**Analysis of Computational Cost.** To provide a comprehensive understanding about the computation cost of our method, we plot the tendency between training time under 200 epoch and the test accuracy as shown in Figure 3, in which we can better observe the computational efficiency as well as the better performance of our approach. To be more specific, we choose ResNet-18 as the basic test model and conduct the experiment about the variation of test accuracy while training to compare the efficacy of different approaches. From Figure 3, we can easily observe the consistent advantage of performance of our approach and comparable training time under the same number of epochs. One interesting point about the "collapse" phenomenon shown in the fourth subplot of Figure 3 reveals the process of this issue. After the learning rate decay around 50-th epoch, the performance of MixUp surprisingly drops steadily to the final result that is even inferior to original ERM. By contrast, our Pani MixUp method achieves consistent improvement on the generalization without the disturbance by any "collapse" issue.

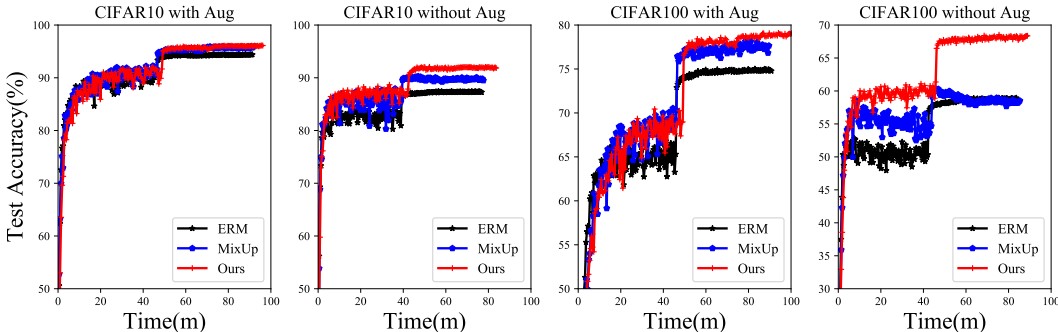

Figure 3: Test accuracy with respect to the training time over ERM, MixUp and our approach. $m$ indicates minutes and the leap in the training comes from the learning rate decay. "with Aug" and "without Aug" denote the settings with and without data augmentation, respectively.

**Further Extension to MixMatch.** To further demonstrate the superiority of our Neighborhood Interpolation MixUp, we embed our approach into MixMatch (Berthelot et al., 2019b), the current state-of-the-art approach that naturally extends MixUp to semi-supervised setting. The resulting approach, Pani MixMatch, elegantly replaces the MixUp part in the MixMatch with our Pani MixUp, thus imposing Pani Mixup by additionally incorporating patch neighborhood correlation knowledge. Results shown in Table 3 demonstrate that Pani MixMatch can further improve the performance of MixMatch in the standard semi-supervised setting, thus verifying the effectiveness and flexibility of our Patch-level Neighborhood Interpolation.

| Methods | CIFAR-10(4,000 labels) |
|---|---|
| PiModel (Laine & Aila, 2016) | $17.41 \pm 0.37$ |
| PseudoLabel (Lee, 2013) | $16.21 \pm 0.11$ |
| Mixup (Zhang et al., 2017) | $13.15 \pm 0.20$ |
| VAT (Miyato et al., 2017) | $11.05 \pm 0.31$ |
| MeanTeacher (Tarvainen & Valpola, 2017) | $10.36 \pm 0.25$ |
| MixMatch (Berthelot et al., 2019b) | $6.24 \pm 0.06$ |
| MixMatch(ours) | $6.26 \pm 0.078$ |
| Pani MixMatch | $\mathbf{6.08 \pm 0.074}$ |

Table 3: Performance of our Pani MixMatch in semi-supervised setting on CIFAR with 4000 labels. The reported result of MixMatch(ours) and Pani MixMatch is under the same random seed, coming from the median of last 20 epoch while training. The results of MixMatch(ours) and Pani MixMatch are the average ones under 4 runs.

## 4 DISCUSSION AND CONCLUSION

The recent tendency of the design of regularization attaches more importance to the consistency and flexibility on various kinds of settings. Along this way, we focus on the proposal of a general regularization motivated by additional leverage of neighboring information existing in the sub-group of samples, e.g., within one batch, which can elegantly extend previous prestigious regularization approaches and generalize well in a wider range of scenarios.

In this paper, we firstly analyze the benefit of leveraging the knowledge from the non-i.i.d relationship while developing more efficient regularization for deep neural networks, thus proposing a general and flexible non-local regularizer called Patch-level Neighborhood Interpolation by interpolating the neighboring patch features in the computation process of network. Furthermore, we customize our Patch-level Neighborhood Interpolation into VAT and MixUp as well as its variant, respectively. Extensive experiments have verified the effectiveness of the two derived approaches, therefore demonstrating the benefit of our Patch-level Neighborhood Interpolation. Our work paves a way toward better understanding and leveraging the knowledge of relationship between samples to design better regularization and improve generalization over a wide range of settings. Since the proposed Pani framework is general and flexible, more regularizations and applications could be considered in the future, such as more traditional regularization methods and the application in natural language processing tasks. Also, the theoretical properties of Pani should also be analyzed.

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
