# OpenReview forum: "Patch-level Neighborhood Interpolation:  A General and Effective Graph-based Regularization Strategy"
_ICLR.cc/2021/Conference — Reject_

### Official Review · AnonReviewer3 · 2020-10-25
**This paper proposes an effective Pani to regularize the networks. The proposed Pani can be applied to input or feature space for better performance. Based on the Pani, it also proposes Pani VAT, Pani MixUP and Pani MixMatch for the generalization performance improvement. The provided results also show the effect of the proposed method.**

**Rating:** 6
**Confidence:** 1

**Review:**

The proposed Pani seems to be novel. It can explore the information of the neighboring relationship between samples and can be regarded as the meta-regularization. For the general formulation of Pani, how does the number of the nearest neighbor patch graphs affect the results?

As I am not familiar with this topic of the paper, there are lots of regularization methods, it would be better to add more details about the regularization methods that do not neighboring relationship among samples.

---

> ### Author Response · Authors · 2020-11-16
> **Response to Reviewer 3**
>
> Thank you for your recommendation for acceptance. Here is our clarification.
>
> $\textbf{1. The sensitivity of the number of nearest neighbor patches.}$
> The exact number of neighbors will be affected by many factors, including patch size, dataset, and the number of layers we conduct our Pani method. Based on our experiments, which are shown in Appendix C, they are normally determined through careful tuning. For instance, we select $K_1$ and $K_2$ as 10 through a linear search in Pani VAT.
>
> $\textbf{2. Related works.}$
> We have provided the preliminary knowledge about VAT and MixUp regularization in Appendix A, both of which do not need neighboring relationship among samples. Please refer to Appendix A for more details.

---

### Official Review · AnonReviewer1 · 2020-10-26
**writing should be improved for better understanding**

**Rating:** 5
**Confidence:** 3

**Review:**

The paper proposes a regularizer called Patch-level Neighborhood Interpolation (Pani). The idea is to construct a graph over patches, and interpolate new patches during training. The paper applies Pani to othter two regularization methods VAT and MixUp. The paper shows better results through experiments.

The writing of the paper can be improved. The current version hurts the merits of the paper, making it not self-explanatory. Here are some questions or weaknesses below, which may be all due to the writing issues.

Section 1: Can the authors define "non-local" and "non-local regularization"? What is the counterpart "local regularization"? From remaining part of the paper, it seems that the neighbhoring patches are from different images, not from the same image.

The paper also motivates from the limitation of iid assumption. But when sampling patches across images, does it also assume iid on the images as it discards other parts of the images?

Figure 1 is very confusing. In the left panel, where are the interpolated patches/images? In the right panel, how to compute $\eta$? How to use lambda to combine three examples ($z_i$, $y_i$) for $i$=1,2,3? Is there a visualization of $r$? Is there a correspondence between images/patches in the left and symbols (e.g., $z$ and $y$) in the right?

Table 1: what are the setups for Pani within VAT? Figure 2 shows multiple design choices, but the paper does not say any values of them in Table 1. The paper is not self-explanatory although it largely claims that it follows some prior work.

Table 1: While it says "without data augmentation", aren't Pani and VAT data augmentation techniques? What happens if all the methods adopt typical data augmentation methods during training (e.g., random flip, random crop, etc.)? Would the proposed method still have advantage?

Figure 2: What is the unit in the y-axis? As there are multiple variables to control (e.g., L, K1, K2, s), what are the fixed values of them in each of the barcharts?

Figure 2: As the paper advocates the graph-based regularization, it must construct the graph. I assume the graph is constructed before-hand, otherwise doing so on-the-fly would increase the compute cost. If this is true, how the proposed method scales to larger datasets (e.g., more images and higher-resolution images)?

Section 3.1: While the authors "argue that the better performance can be expected if we construct perturbations on more hidden layers despite the increase of computation", there is no justification on the argument. The paper should conduct such an experiment to justify this by applying Pani on more layers given that the cost is low as studied in Figure 2.

Section 3.2: What is "mask mechanism"? The paper does not explain it.

Section 3.2: What does it mean by "one kind of collapse"? Although Figure 3 illustrate the issue in the fourth plot, can the authors expand the discussion? What is the learning rates? Does learning rate scheduler cause the issue? What other explanations? Does using another learning rate resolve the issue?

Table 3: What is the rationale behind reporting results "coming from the median of last 20 epoch while training"? What are the total training epochs of each methods?

The appendix is not attached to the paper.


----------
post-rebuttal
----------
I appreciate that authors have provided rebuttal that addresses some of my questions. I've read the updated paper and other reviewers' comments. In general, I'd like to maintain my initial rating as a borderline paper. Here are two main reasons.

First, I realize that authors are not familiar with the policy, because they did not attach the appendix to the manuscript but uploaded as a separate file. As a result, their updated paper did not address issues in the review effectively. Authors simply say something like below

- "We promise to improve our writing based on your suggestions in the revised version."
- "Due to the space limit, we put our related experimental details into Appendix C." (9 pages are allowed by policy)
- "We will supplementary the results of Pani Mixup(+hidden) to justify it if accepted."

Second, in terms of other data augmentation, the authors merely say "As data augmentation is a common trick, consistent improvement can be easily expected across all methods". I don't know if this is true unless there is a justification.

---

> ### Author Response · Authors · 2020-11-16
> **Response to Reviewer 1**
>
> Thank you for your constructive feedback. Here is our clarification for all your questions. Also, we have provided our appendix in the supplementary materials in submission and please refer to it for more details mentioned below.
>
> $\textbf{1. Local and non-local regularization.}$
> For image classification, non-local regularization is derived from non-local image filtering, which we discussed in Appendix B. Local regularization typically refers to the regularization that is constructed based on i.i.d. data, including the great majority of regularization strategies, such as VAT. By contrast, non-local regularization tends to leverage the information from relationships between images, patches or even pixels to design strategies that can benefit the task. Please refer to Appendix B for more details.
>
> $\textbf{2. I.i.d. assumption.}$
> The i.i.d assumption is implicitly made while deriving a flurry of machine learning methods, such as SVM and BP. By contrast, while constructing the candidate peer images, we apply random matching or computing the semantically nearest image neighbors, which is different from the i.i.d. assumption we mentioned before.
>
> $\textbf{3. Figure 1}$
> For the left part of Figure 1, for each resulting patch, it was constructed by linear interpolation by neighboring patches (with the same color). For the right part of Figure 2, the interpolation coefficient $\eta$ or $\lambda$ are computed by optimizing Eq.2 in PaniVAT and sampling based on Eq.4 in Pani MixUp, respectively. Figure 2 presents the general formulation of our Pani method.
>
> $\textbf{4. Table 1: Setup about Pani VAT.}$
> Due to the space limit, we put our related experimental details into Appendix C, as stated in the Implementation Details in Section 3.1. Also, we have provided some preliminary knowledge about VAT, MixUp in Appendix A to help readers get a better understanding of our work.
>
> $\textbf{5. Table 1: Setting without data augmentation.}$
> Setting without data augmentation is the default one in the semi-supervised setting to conduct a comparison. As data augmentation is a common trick, consistent improvement can be easily expected across all methods. Thus, in our paper we focus on the normal setting without data augmentation.
>
> $\textbf{6. Figure 2: some details.}$
> The unit of y-axis is second. The fixed hyper-parameters are what the Pani VAT employed in Table 1. We have provided the details of hyper-parameters in Appendix C.
>
> $\textbf{7. Figure 2: the graph construction.}$
> The graph is constructed within each batch during training, which followed PeerNet[1]. As our K-NN graph is patch-level, it is more computationally efficient compared with the pixel-level graph used in PeerNet. More importantly, we can adjust the number of peer images $K_1$ and patch neighbors $K_2$, which can help us scale to larger datasets while balancing the effectiveness and computational cost.
>
> $\textbf{8. Section 3.1.}$
> Section 3.1: Firstly, Figure 2 shows that as the number of layers increases, the computational cost will also increase almost linearly. In addition, we have verified that Pani VAT(+hidden) can achieve better improvement than Pani VAT(input) version. Further, as Mixup is only on the input layer, we focus on Pani MixUp(input) for a fair comparison. We will supplementary the results of Pani Mixup(+hidden) to justify it if accepted.
>
> $\textbf{9. Section 3.2: mask mechanism.}$
> Section 3.2: The mask mechanism can be viewed as dropout or enforcing sparsity on the interpolation coefficient $\eta$, which can help to reduce redundant information while conducting Pani MixUp. Please refer to the explanation about Mask Mechanism in Appendix C Implementation Details.
>
> $\textbf{10. Section 3.2: collapse.}$
> Section 3.2: We strictly followed the released code, including the default learning rate and scheduler, from MixUP, but we found that MixUp cannot achieve consistent improvement on CIFAR-100 without the data augmentation. We argue that MixUP is not very effective without data augmentation as it is intuitive that mixing augmented images can provide more informative supervised signals. By contrast, our Pani method is patch-level, which is more flexible to capture the delicate supervised signals. Thus, our Pani method is not highly impacted under whether the data augmentation is applied or not, and thus does not suffer from the “collapse” issue.
>
> $\textbf{11. Table 3.}$
> Table 3: We strictly followed the released code from MixMatch and employed the same epoch while training. The median of the last 20 checkpoints’ error rate also followed MixMatch to mitigate the randomness.
>
> We have provided our appendix in the supplementary materials in submission. We promise to improve our writing based on your suggestions in the revised version.
>
> [1] Jan Svoboda, Jonathan Masci, Federico Monti, Michael M Bronstein, and Leonidas Guibas. Peernets: Exploiting peer wisdom against adversarial attacks. ICLR, 2018.

---

### Official Review · AnonReviewer2 · 2020-10-28
**Review of "Patch-level Neighborhood Interpolation: A General and Effective Graph-based Regularization Strategy"**

**Rating:** 6
**Confidence:** 2

**Review:**

The paper proposes a general regularizer called the Patch-level Neighborhood Interpolation (Pani) that constructs patch-level graphs at different levels of neural networks. Specifically, it is based on the k-nearest patch neighbors at each layer and linear interpolation for each patch. By applying this proposed regularizer framework into two special cases and get Pani VAT and Pani MixUp. Numerical experiments are comprehensive and convincing.

Pros:
A new type of regularization for neural networks is proposed.
Two special Pani based algorithms within a batch are proposed with applications in image classification.
Experimental performance in terms of accuracy and running time shows that the Pani regularizer can improve the algorithm.

Cons:
The motivation of combining patch based k-NN and linear interpolation is not fully clear. Can we replace each by one of other related methods?
No theoretical guarantees for this improvement are provided, which could be strengthened in the revision.
How can the Pani be integrated to other types of machine learning regularizations? Some practical guidance could be provided.

Overall, the work is important and interesting which could provide insights to other related works in the area.

---

> ### Author Response · Authors · 2020-11-16
> **Response to Reviewer 2**
>
> Thank you for your recommendation for acceptance. Here is our clarification.
>
> $\textbf{1. Motivation.}$
> (1) Patch-level: As stated in the first paragraph in Section 2, our patch-level method can achieve a computational advantage over the pixel-level regularization. Also, a flexible patch size can be chosen according to the size of the receptive field.
> (2) K-NN graph: When constructing the neighboring relationship, the K-NN graph is preferred since it has a computational advantage. Similarly, the related work PeerNet [1] also constructed a K-NN graph to develop a regularization, but they designed it to defend against adversarial attacks, which is different from our setting.
> (3) Linear interpolation is a simple, natural yet effective way to construct the neighboring relationship, which has a huge computational advantage over the GAT-based non-linear form proposed in PeerNet [1].
>
> $\textbf{2. Theoretical guarantee.}$
> We have to admit that it is tough to conduct the theoretical analysis for the graph-based representation methods due to the non-i.i.d. property. To make up for this shortage, we conduct extensive experiments on CIFAR-10, CIFAR-100 and Tiny-ImageNet across VAT, MixUP and MixMatch methods to verify the effectiveness of our Pani method. We leave the theoretical guarantee of our approach as future work.
>
> $\textbf{3. Further integration.}$
> As stated in Introduction, we find that among all the explicit and implicit regularizations, regularization with stochastic transformation, perturbations and randomness, such as VAT, MixUp and their variants, play a key role in the deep learning models. We argue that other regularization with stochastic transformation and perturbations, like VAT, MixUP, has a similar philosophy to incorporate Pani methods. Huge potential can be further explored in the future, and we also leave this integration as future work.
>
> [1] Jan Svoboda, Jonathan Masci, Federico Monti, Michael M Bronstein, and Leonidas Guibas. Peernets: Exploiting peer wisdom against adversarial attacks. ICLR, 2018.

---

### Official Review · AnonReviewer4 · 2020-10-28
**This paper presents a new regularized method via Patch based interpolation.  But several weaknesses lead me to reject this paper.**

**Rating:** 5
**Confidence:** 4

**Review:**

This paper proposed a new regularization method via patch level interpolation.  During the training,  images within a batch will be used to construct an image graph. For example, for a certain image, its nearest neighbors in the feature spaces will be used.  Then patches from its neighbors will be used to interpolate to each patch in that given image.  Thus a straightforward application for such regularization is semi-supervised training.  Moreover, in this paper it has demonstrated such regularization can be extended with virtual adversarial training and mixup training.

Although the proposed training strategy is simple, and the proposed patch interpolation is able to achieve better performance when compared with some baselines. It is still not comparable to existing semi-supervised training works such as Mixmatch or FixMatch. In the last section of the paper, the author shows the extension with mixmatch  by surpass the mixmatch with little improvement. However, when compared to Fixmatch, the performance gap becomes larger. (Note, fixmatch seems simpler than proposed method in terms of computational cost)

Thus the proposed method does not convinced me for its effectiveness.

missing reference.
[1]FixMatch: Simplifying Semi-Supervised Learning with Consistency and Confidence

---

> ### Author Response · Authors · 2020-11-16
> **Response to Reviewer 4**
>
> Thank you for your constructive feedback. Here is our clarification.
>
> The state-of-the-art performance of FixMatch does not contradict with the effectiveness of our Pani method as the generality is what our Pani method values, which can enhance existing regularization methods, including VAT, MixUp, MixMatch and FixMatch. We demonstrate that the Pani method is a general, simple yet effective off-the-shelf strategy in both the semi- and supervised setting. Additionally, FixMatch is closely related to MixMatch, both of which are based on consistency regularization and pseudo-labeling. As we have demonstrated the improvement of Pani-MixMatch compared with MixMatch in Table 3, the similar improvement can be expected in the Pani version of FixMatch, in which the implementation of Pani on FixMatch is similar to Pani-MixMatch. Lastly, VAT, Mixup and Mixmatch are more representative regularization than FixMatch, which is still an unpublished work based on our knowledge. To better verify the effectiveness of our Pani method, we focus on the former three representative strategies.

---

### Decision · Program_Chairs · 2021-01-07
**Final Decision**

**Decision:**

Reject

**Comment:**

This paper proposes a novel way (Pani) that constructs image patch-level graphs and then linearly interpolates the patch-level features. The authors show how this can be used in Virtual Adversarial Training (PaniVAT) and Mixup/MixMatch (Pani Mixup). The method is shown to improve classification compared to standard VAT and related techniques on CIFAR-10 (low data setting), as well as outperform Mixup on CIFAR-10/CIFAR-100/TinyImageNet (standard setting, multiple different architectures) with and without data augmentation.

Reviewer 4 liked that the method was simple, but was not convinced of its effectiveness because of the baselines that were chosen. Specifically they thought that FixMatch was a stronger baseline than MixMatch. The authors said that Pani is complementary to FixMatch and similar improvement could be expected when applying Pani to FixMatch instead of MixMatch.

Reviewer 2 appreciated that the work was “important and interesting” and noted that the experiments showed that Pani improved existing algorithms. They were concerned with lack of motivation and lack of theoretical guarantees. The authors clarified motivation in their response to the reviewer but, understandably, were unable to provide any theoretical analysis.

Reviewer 1 expressed disappointment with the writing and understandability of the paper. I read the paper myself and I agree. They posed several clarifying questions to the authors, to which the authors responded. I note that the reviewer could not find the appendix, but it was attached separately as supplementary material.

Reviewer 3 wrote a very short review and stated that they are not familiar with the topic of the paper. With three other full reviews, I have discounted R3’s review because of their extremely low confidence. They also asked a couple of clarifying questions, to which the authors responded. I found the authors’ response satisfying.

Overall, two reviewers are not extremely excited about the paper and one reviewer thinks the work is interesting but has concerns about clarity. I think that overall it is a neat idea, but the paper could use more polishing and clarification. Compared to other borderline papers in my stack, it is not over the bar. It could get there with further work. I hope the authors continue to improve the paper and re-submit it in the near future.